

# Is the future already here? The impact of climate change on the distribution of the eastern coral snake (*Micrurus fulvius*)

Jennifer N. Archis[1], Christopher Akcali[2], Bryan L. Stuart[3], David Kikuchi[4] and Amanda J. Chunco[1]

[1] Department of Environmental Studies, Elon Univeristy, Elon, NC, United States of America
[2] Biology Department, University of North Carolina, Chapel Hill, NC, United States of America
[3] North Carolina Museum of Natural Sciences, Raleigh, NC, United States of America
[4] Center for Insect Science, University of Arizona, Tucson, AZ, United States of America

Corresponding author
Amanda J. Chunco,
achunco@elon.edu

## ABSTRACT

Anthropogenic climate change is a significant global driver of species distribution change. Although many species have undergone range expansion at their poleward limits, data on several taxonomic groups are still lacking. A common method for studying range shifts is using species distribution models to evaluate current, and predict future, distributions. Notably, many sources of 'current' climate data used in species distribution modeling use the years 1950–2000 to calculate climatic averages. However, this does not account for recent (post 2000) climate change. This study examines the influence of climate change on the eastern coral snake (*Micrurus fulvius*). Specifically, we: (1) identified the current range and suitable environment of *M. fulvius* in the Southeastern United States, (2) investigated the potential impacts of climate change on the distribution of *M. fulvius*, and (3) evaluated the utility of future models in predicting recent (2001–2015) records. We used the species distribution modeling program Maxent and compared both current (1950–2000) and future (2050) climate conditions. Future climate models showed a shift in the distribution of suitable habitat across a significant portion of the range; however, results also suggest that much of the Southeastern United States will be outside the range of current conditions, suggesting that there may be no-analog environments in the future. Most strikingly, future models were more effective than the current models at predicting recent records, suggesting that range shifts may already be occurring. These results have implications for both *M. fulvius* and its Batesian mimics. More broadly, we recommend future Maxent studies consider using future climate data along with current data to better estimate the current distribution.

# INTRODUCTION

Climate change is dramatically altering the distribution of species across the globe (*Parmesan, 2006*; *Chen et al., 2011*). Climate change has been hypothesized to drive range shifts at poleward range limits, especially for warm-adapted species (*Parmesan, 2006*).
*Hughes (2000)* notes that a change of 3 degrees Celsius can result in poleward isotherm shifts of 300–400 km in temperate zones; the ranges of species are expected to expand in response. Indeed, in a review of range shifts in 434 species of varied taxa including birds, butterflies, and alpine herbs, 80% had moved in the direction predicted by climate change (*Parmesan & Yohe, 2003*). The cascading effects of range shifts are manifold and include: decoupling of intricate predator–prey interactions and multitrophic effects (*Gilman et al., 2010*), increased invasion (*Hellmann et al., 2008*) and infection (*Pounds et al., 2006*), increased intraspecific competition (*Huey et al., 2009*), and potential competitive exclusion in extreme cases (*Gilman et al., 2010*; *Tylianakis et al., 2008*; *Chapin III et al., 2000*).

The ranges of ectothermic taxa may be especially sensitive to the effects of climate change. In ectotherms, body temperature is either closely linked to environmental conditions, i.e., poikilothemy, or is regulated behaviorally (*Huey, 1982*). Thus, ectotherms must rely on particular climate conditions to maintain proper metabolic function (*Currie, 2001*; *Hansen et al., 2001*) and to optimize reproductive physiology (*Girons, 1982*). Both mean temperature and precipitation, as well as variation in temperature and precipitation, have been shown to influence performance in ectotherms (*Clusellas-Trullas, Blackburn & Chown, 2011*). Because of this strong reliance on temperature and the reduced energy costs for thermoregulation predicted to occur as a result of climate change in many temperate areas, it has been hypothesized that many reptile species will show an increase in range size (*Currie, 2001*; *Hansen et al., 2001*).

Despite the importance of environmental conditions on physiological performance in reptiles, relatively few studies have assessed the impact of climate change on reptile species. For example, two major reviews on distributional shifts (*Parmesan & Yohe, 2003*; *Parmesan, 2006*) included only two reptile species, both lizards. An additional meta-analysis (*Chen et al., 2011*) only included a single paper on reptiles and amphibians, and that study was geographically restricted to Madagascar. Thus, snakes provide a unique opportunity to examine the generality of patterns in range shifts for ectotherms.

One exceptionally understudied species is the eastern coral snake (*Micrurus fulvius*). Although coral snakes are well-known for their striking coloration (Fig. 1) and potent venom, relatively little is known about the natural history of this species (*Steen et al., 2015*; *Jackson & Franz, 1981*). *M. fulvius* has historically ranged throughout the Southeastern United States from the southern tip of Florida to the Sandhills of North Carolina (Fig. 2). At a coarse scale, *Micrurus fulvius* inhabits sandy flatwoods and maritime forests (*Beane et al., 2010*), and one recent survey suggest a microhabitat preference for sandy soils and scrub/shrub habitat (*Steen et al., 2015*). Their diet primarily consists of other snakes and lizards (*Beane et al., 2010*). Although the International Union for the Conservation of Nature (IUCN) listed *M. fulvius* as "Least Concern" in 2007 based on its total global population size (*Hammerson, 2007*), it is of significant conservation concern at the local level throughout most of its range; it is listed as Endangered in North Carolina (*North Carolina Wildlife Resources Commission, 2014*), Imperiled in South Carolina (*South Carolina Department of Natural Resources, 2014*), and of Highest Conservation Concern in Alabama (*Outdoor Alabama, 2017*). The fate of these populations at the edges of its range depends on how its range shifts in response to climate change.

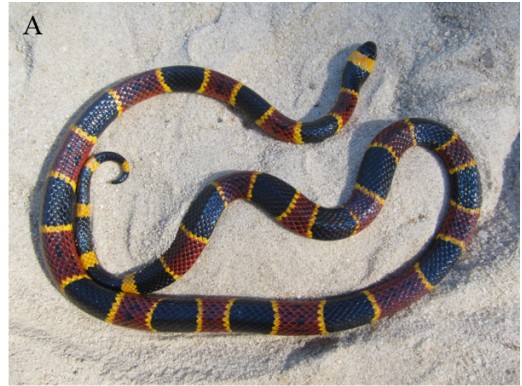

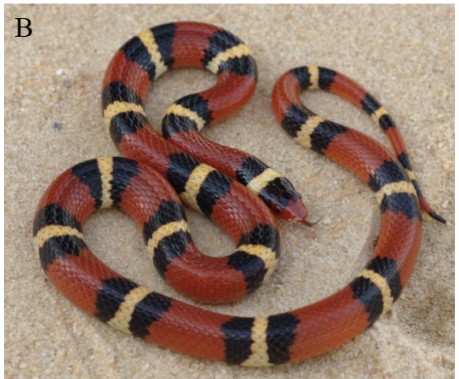

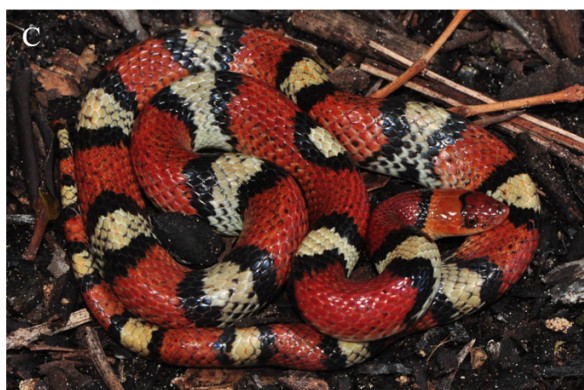

**Figure 1** Pictures of (A) the venomous coral snake (*M. fulvius*) (photo credit: Christopher Akcali) and its non-venomous mimics, (B) the scarlet king snake (*Lampropeltis elapsoides*) (photo credit: David Kikuchi) and (C) the scarlet snake (*Cemophora coccinea*) (photo credit: Troy Hibbits).

Species distribution modeling (SDM) is an increasingly important methodology for predicting range shifts resulting from anthropogenic climate change. This approach integrates species occurrence data with environmental variables to predict suitable environments (*Pearson & Dawson, 2003*; *Graham et al., 2004*; *Fitzpatrick et al., 2008*). Species Distribution Models have also been used to predict future range shifts due to climate change by creating a SDM using current climate conditions and projecting that model onto future predicted climate conditions (*Thomas et al., 2004*; *Araújo et al., 2005*; *Guisan & Thuiller, 2005*; *Hole et al., 2011*).

One of the most common datasets used in SDMs is WorldClim, which provides interpolated bioclimatic layers based on weather station data collected between 1950–2000 (*Hijmans et al., 2005*). This dataset has been cited more than 10,100 times per Google Scholar as of June 1, 2017. Yet, the magnitude of climate change that has taken place since 2000 (*Karl et al., 2015*) may have already rendered these 'current' climate conditions less accurate at predicting species distributions than 'future' climate predictions.

Here, we explore the impact of climate change on the distribution of *M. fulvius* by: (1) identifying the current range and suitable environments of *M. fulvius* in the Southeastern United States, (2) predicting future shifts in the range of *M. fulvius* under different climate

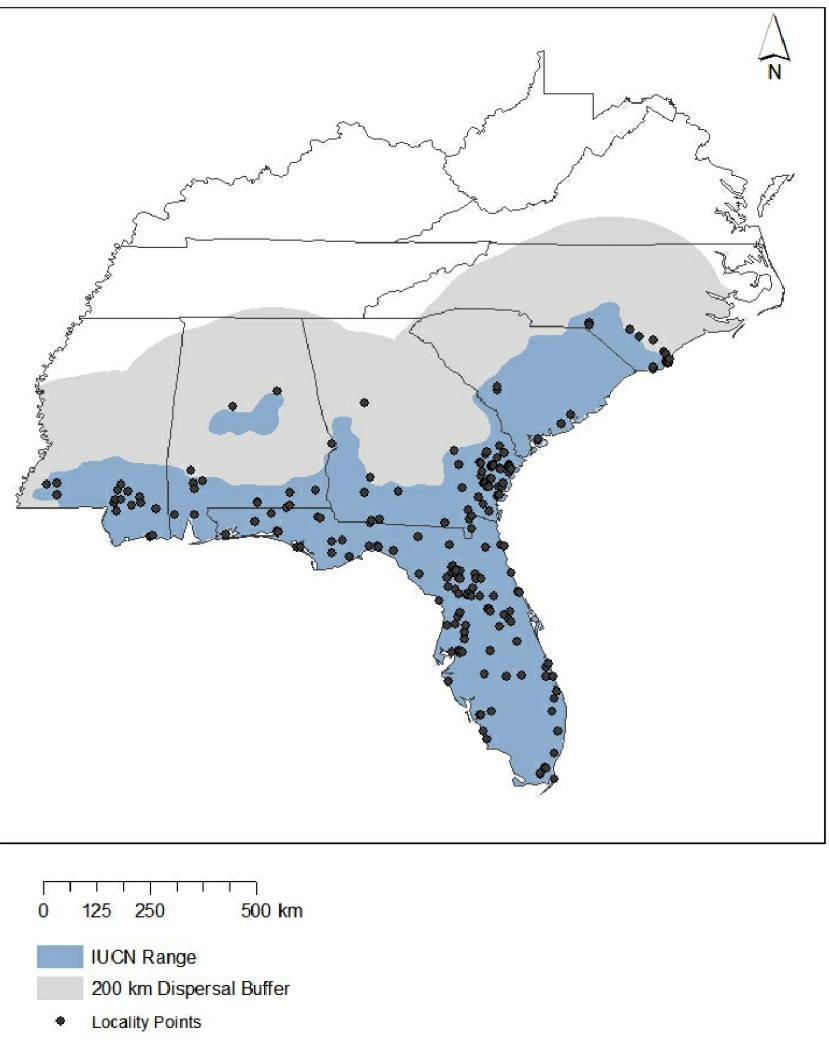

**Figure 2  Study area.** Blue represents range of the eastern coral snake according to the International Union for the Conservation of Nature, while gray indicates the full area of study used in this research when accounting for a 200-km dispersal buffer. Points shown are occurrence records ($n = 142$) used to create all models. Note that ten points did not overlap with bioclim data and therefore were not used in models.

change scenarios, and (3) comparing the effectiveness of both current and future models in evaluating recent coral snake occurrences in the years between 2001–2015.

## METHODS

### Species distribution modeling—current climate

The ecological niche program Maxent version 3.3.3k (*Phillips, Dudík & Schapire, 2004*) was used to create a correlative species distribution model, executed through the Java Applet. Maxent uses environmental data from locations where the species of interest has been observed to create a map of habitat suitability across the area of interest. Maxent was chosen over competing modeling approaches because it has been well validated for use with

presence-only data and is capable of dealing with complex interactions between response and predictor variables (*Elith et al., 2006*) while remaining robust to small sample sizes (*Wisz et al., 2008*). Presence-only methods, which do not require costly and difficult-to-collect absence data (*Gu & Swihart, 2004*), can be used to model the same environmental relationships as presence-absence methods provided that biases are accounted for (*Elith et al., 2011*). This is especially useful for *M. fulvius* because, as is the case for most snake species, they are difficult to detect due to their limited active periods and cryptic behavior (*Christoffel & Lepcyzk, 2012*; *Guimaraes et al., 2014*). In addition, snake species are often patchily distributed and have low population sizes (*Segura et al., 2007*) making it difficult to obtain robust absence data.

Locality information was provided by herpetology collections from museums throughout the United States. These data were retrieved from multiple sources: the HerpNet2Portal (http://www.HerpNet2.org, accessed 2012-05-30; note that HerpNet has been folded into VertNet and specimen information is now available through http://portal.vertnet.org/search); the Global Biodiversity Information Facility (GBIF) (http://www.gbif.org, accessed 2013-06-26); iDigBio (http://www.idigbio.org, accessed 2016-05-23); and some locality points were provided directly from museum curators for collections that had not yet been digitized with any of the aforementioned databases. The search term "*Micrurus fulvius*", was used with no other qualifiers. Duplicate points were removed as were localities found west of the Mississippi River (as those specimens are now recognized as *Micrurus tener*), leaving a total of 1,074 unique records collected from 35 different institutions (see the specific museum collections listed in Appendix S1). All points were georeferenced and the precision of the locality data was estimated according to best practices proposed by *Chapman & Wieczorek (2006)*. Records outside the timeframe of the climate data (between 1950–2000) were removed (leaving $n = 747$). Only points with an uncertainty less than or equal to 5 km were retained for the model, which reduced the number of records to 242. This reduced number of records provided substantial geographic coverage of the entire range, with the exception of a small gap in both central and western Florida. Because there were a high number of records from within the 1950–2000 timeframe, despite many of them not meeting our stringent requirements for spatial accuracy, we included seven additional localities from these regions to provide complete coverage of the range, leaving a total of $n = 249$ records. Preliminary models showed using occurrence data with lower uncertainty (i.e., less than 3 km) greatly reduced the number of locality points without improving model performance. Previous studies have also shown that model performance is only minimally impacted at uncertainty thresholds below 5 km (*Graham et al., 2008*).

Datasets that are not collected via systematic sampling are often subject to biases based on accessibility (*Kadmon, Farber & Danin, 2004*), dispersal ability, and level of scientific and community interest (*Fourcade et al., 2014*). While we used some of the most robust available data for an elusive snake species (*Franklin et al., 2009*), bias can be reduced through the background selection process previously discussed. In addition to background selection, *Fourcade et al. (2014)* note that spatial filtering will further reduce bias associated with non-systematic sampling. Therefore, the *Spatially Rarefy Occurrence Data for SDMs*

tool set within the SDMtoolbox (http://sdmtoolbox.org/) was used to classify and spatially filter the climate variables into different homogenous areas and reduce the number of points clustered around these homogenous zones (*Brown, Bennett & French, 2017*). SDMtoolbox is an ArcGIS extension that includes several tools to assist in running species distribution models. This further reduced the number of points to 142. Ten of these points were found to be outside of the range of one or more environmental variable layers used because they were located on islands that did not have bioclimate data available. Therefore, Maxent did not use these points in the model, reducing the number of used occurrence points to 132. This number is well above the locality sample size at which Maxent has been shown to be effective (*Hernandez et al., 2006*). In addition, the benefits of additional occurrence points appear to plateau after 50 (*Hernandez et al., 2006*); however, some studies disagree with 50 as either too conservative (*Jiménez-Valverde & Lobo, 2007*) or too generous (*Proosdij et al., 2016*). In either case, the number of points we used was well above this threshold.

For environmental data, we used 19 continuous bioclimatic variables downloaded from WorldClim 1.4; these climate layers were generated using weather station data compiled between 1950 and 2000 (*Hijmans et al., 2005*; http://www.worldclim.org; Table 1; Appendix S2). These data are commonly used in Maxent modeling (as noted above), and provide temperature and precipitation metrics that are important abiotic factors in driving the distribution of ecotherms. Additionally, as coral snakes have been shown to have a preference for specific soil types (*Steen et al., 2015*) we used categorical soil type data from the Harmonized World Soil Database created by the Food and Agriculture Organization of the United Nations (FAO) and the International Institute for Applied Systems Analysis (IIASA) (*FAO/IIASA/ISRIC/ISSCAS/JRC, 2012*) (Table 1). We included the same soil layer in both current and future climate models as soils can be expected to remain stable over these short time frames. Raster grids for the bioclimatic variables were at a spatial resolution of 2.5 arc-minutes (less than a 5 km$^2$ resolution), and soil type data were scaled to match. This resolution was chosen because it is close to the accuracy of the occurrence locality data that were used, thereby maximizing the likelihood that conditions at the point of collection are correctly portrayed while also maintaining the high resolution of the data (*Anderson & Raza, 2010*).

The extent of the study area can also heavily influence model output (*Elith, Kearney & Phillips, 2010*). Large study regions can often lead to overfitting (*Anderson & Raza, 2010*), which lowers model transferability across time (*Araújo & Rahbek, 2006*). This is because Maxent uses 'pseudoabsence' points drawn randomly from throughout the study area and evaluates the model by comparing the environment at points where the species is known to be present with these pseudoabsence points (*Phillips, Anderson & Schapire, 2006*). Restricting the study extent to areas where the species being modeled could potentially occur, as opposed to areas far outside the range, restricts pseudoabsence points to areas where they are informative (*Barbet-Massin et al., 2012*). Thus, all 20 environmental variables were trimmed using ArcGIS 10.3 to include only those areas that overlap with the historical range of *M. fulvius* (*NatureServe, International Union for Conservation of Nature, 2007*), with a buffer of 200 km around the known range. This buffer accomplishes two goals. First, the buffered region includes all the current habitat where *M. fulvius* have been

 

**Table 1  A description of the environmental data used in the four Maxent model types.** Values shown are percent contributions in each model, averaged across all runs under current climate conditions. Maps of each variable across the study area are provided in Appendix S2, and the response curves for each variable in the full model are provided in Appendix S3.

| Variable | Full model | Reduced model | Moderate correlation ($r < 0.7$) | Low correlation ($r < 0.5$) |
|---|---|---|---|---|
| Annual mean temperature | 4.2 | | 64.8 | 73.9 |
| Mean diurnal range in temperature | 2.1 | | 3.4 | |
| Isothermality | 1 | | | |
| Temperature seasonality | 7.3 | 11 | | |
| Max temperature of warmest month | 1.6 | | 2.1 | |
| Min temperature of coldest month | 0.3 | | | |
| Temperature annual range | 0.9 | | | |
| Mean temperature of wettest quarter | 2 | | 9 | 11.9 |
| Mean temperature of driest quarter | 0.8 | | 2.3 | |
| Mean temperature of warmest quarter | 0.2 | | | |
| Mean temperature of coldest quarter | 26 | 34.1 | | |
| Annual precipitation | 1.1 | | 3.2 | |
| Precipitation of wettest month | 1.8 | | | |
| Precipitation of driest month | 6.7 | 6.8 | | |
| Precipitation seasonality | 2.3 | | | |
| Precipitation of wettest quarter | 3.1 | | | 3.4 |
| Precipitation of driest quarter | 1.5 | | 5.4 | |
| Precipitation of warmest quarter | 27.8 | 39.6 | | |
| Precipitation of coldest quarter | 2.3 | | | |
| Soil type | 6.9 | 8.6 | 9.9 | 10.8 |

observed (including seven known localities that occurred as far as 45 km outside of the range identified by the IUCN). This ensured that all suitable habitat during the 1950–2000 time frame was used in the model. Second, including an additional buffer beyond the known range captures all the potential area of future suitable habitat (Fig. 2). This method of background selection produces more realistic predictions than large study regions because it removes areas where species are unable to disperse (*Barve et al., 2011*; *Anderson & Raza, 2010*).

Because the choice of environmental variables can influence model output, several independent models were run with different compositions of environmental variables (Table 1). Four strategies were used to select variables for each model. The first model was comprised of the full suite of variables, both bioclimatic and soil type (the Full model). However, many of the variables included in the Full model may have minimal influence and cause model overfitting (*Baldwin, 2009*). Therefore, the results of the first model were used to identify variables with a significant (>5%) contribution; these variables were used in the second model (the Reduced model). Finally, because of the potential for highly correlated variables, subsets of bioclimatic variables within 2 different correlation thresholds— $r < 0.7$ (Moderate Correlation model) and $r < 0.5$ (Low Correlation model)—were identified using

SDMtoolbox. The correlation threshold of 0.7 is commonly used in niche modeling and has been well validated (*Dormann et al., 2013*). We also used 0.5 to determine if a more rigorous threshold influenced model output.

Maxent's logistic output settings return a raster grid where each cell is assigned a value between 0 and 1, with a value of 0 representing a low probability of species occurrence and therefore low habitat suitability and a value of 1 representing a high probability of species occurrence and therefore high habitat suitability (*Phillips & Dudík, 2008*). The logistic output was used in Maxent because it is easier to interpret than raw and cumulative output formats (*Phillips & Dudík, 2008*). The regularization multiplier was set to 1.0, maximum iterations 500, and all models were replicated using 10-fold cross-validation. In cross-validation, species locality data is divided randomly into a number of partitions of equal size (k); models are then trained iteratively using k-1 partitions, while the final partition is retained for model evaluation (*Merow, Smith & Silander, 2013*; *Radosavljevic & Anderson, 2013*). Specifically, 10-fold cross-validation is a commonly used metric in the literature (e.g., *Elith et al., 2011*; *Kearney, Wintle & Porter, 2010*). Jack-knifing was used to determine variable importance. The averages of these runs were used in all analyses. Results were visualized in ArcMap 10.3 using a WGS84 projection.

Two primary methods were used to assess model fit: area under the receiver operating characteristic (ROC) curve plots and true skill statistic (TSS). Area under the curve (AUC) is one of the most common statistics used for model evaluation. Although this metric has been criticized (*Lobo, Jiménez-Valverde & Real, 2008*), it is considered reliable enough for comparing models of a single species in the same area with the same predictor variables (*Fourcade et al., 2014*). True skill statistic is a threshold-dependent evaluation method that is based off of Cohen's Kappa. The Kappa statistic is used in presence/absence models to calculate model accuracy normalized by the accuracy predicted by random chance (*Allouche, Tsoar & Kadmon, 2006*). TSS remains independent of prevalence and is therefore considered a robust test for validation (*Allouche, Tsoar & Kadmon, 2006*).

## Species distribution modeling—future climate

Data for future climate predictions were taken from the Consultative Group for International Agricultural Research (CGIAR) research program on Climate Change, Agriculture and Food Security (CCAFS). These raster grids were downloaded at an identical resolution to the current data (2.5 arc-minutes). Soil type data are assumed to remain stable through time even under the influence of climate change; because of this, soil type data used for current models (taken from the FAO/IIASA Harmonized World Soil Database) were used along with the projected future climate variables to comprise a full suite of variables.

Global Climate Models, or GCMs, represent the diversity of pathways that the future climate may follow in coming years. Each GCM comes from a different parent agency and represents a different scenario based on the chosen Representative Concentration Pathway (RCP) and model inputs. RCP is an emissions scenario that quantifies a potential climate future by projecting greenhouse gas trajectories (*IPCC, 2014*). Smaller RCPs such as 2.6 have drastically lower projected greenhouse gas concentrations than larger RCPs such as

8.5. We selected an RCP of 4.5, which represents a moderate greenhouse gas mitigation scenario (*Kopp et al., 2014*). Under this scenario, the global mean surface temperature is likely to increase between 1.1 °C to 2.6 °C relative to the 1986–2005 period by the end of the century (*IPCC, 2014*). This RCP assumes moderate global efforts to reduce greenhouse gas emissions. If these reductions in emissions do not occur, this RCP may underestimate future climate warming (*Kopp et al., 2014*).

Each Coupled Model Intercomparison Project (CMIP5) model configuration is the product of a unique group of variables and therefore has its own benefits. Best practice is to utilize multiple models because: (1) they allow a range of possible outcomes to be evaluated and (2) similar outcomes from different models increase confidence (*Collins et al., 2013*). Therefore, two CMIP5 configurations were selected: MIROC-ESM, a cooperative effort by the University of Tokyo's Atmosphere and Ocean Research Institute, National Institute for Environmental Studies, and Japan Agency for Marine-Earth Science and Technology, and GISS-E2-R (NINT) from the Goddard Institute for Space Studies. Both models used the same RCP as mentioned above.

Both CMIP5 configurations have different coupling compilations that result in different equilibrium climate sensitivity (ECS) and transient climate response (TCR) values: 4.7 ECS and 2.2 TCR for MIROC-ESM (*Andrews et al., 2012*) and 2.1 ECS and 1.5 TCR for GISS-E2-R (NINT) (*Flato et al., 2013*). Equilibrium climate sensitivity is a measure of the change in equilibrium global mean surface temperature after a doubling of atmospheric $CO_2$ concentration, while transient climate response is a measure of expected warming at a given time (*Collins et al., 2013*). Most models agree on a range for these values between 1.5 and 4.5 for ECS, and between 1 and 2.5 for TCR (*Collins et al., 2013*). Because MIROC-ESM is above this range, it projects a dramatic change in mean surface temperature and is therefore considered more pessimistic than GISS-E2-R (NINT).

In addition to the resulting suitability maps, the output of multivariate environmental similarity surfaces (MESS) analysis was examined (*Elith, Kearney & Phillips, 2010*). MESS analysis compares the environmental similarity of variables and identifies areas where one or more environmental variables are outside of the training range and were implemented with the Maxent Java Applet. Negative values indicate a novel climate, and the magnitude indicates the degree to which a point is out of range from its predictors. Positive values indicate climate similarity and are scored out of 100, with a score of 100 indicating that a value is entirely non-novel (*Elith, Kearney & Phillips, 2010*).

## Model change

To quantify model change, results were first converted from continuous to binary prediction. This step required the use of a threshold cutoff that defined areas with a suitability greater than the threshold as "good" habitat and areas with a suitability less than the threshold as "poor" habitat. For this study, the maximum training sum of sensitivity and specificity (Max SSS) threshold was chosen as it has been found to be a strong method for selecting thresholds with presence-only data (*Liu, White & Newell, 2013*). This threshold aims to maximize the summation of both sensitivity (true predicted presence) and specificity (true predicted absence). It is considered a strong choice when

**Table 2** Results of model analysis (max SSS threshold, mean area under the curve, and true skill statistic) for each model type.

| Model type | Threshold | Average test AUC | TSS |
|---|---|---|---|
| Full model | 0.3378 | 0.8280 ± 0.0509 | 0.6314 |
| Reduced model | 0.3293 | 0.8283 ± 0.0470 | 0.5831 |
| Moderate correlation ($r < 0.7$) | 0.3079 | 0.8214 ± 0.0487 | 0.5733 |
| Low correlation ($r < 0.5$) | 0.2662 | 0.8148 ± 0.0494 | 0.5398 |

modeling for conservation purposes because the costs of omission (false negative) are greater than those of commission (false positive) (*Jiménez-Valverde & Lobo, 2007*). Binary change was quantified by subtracting the recent model from the extrapolated future model. The resulting raster displays a score of either 0, 1, or −1: 0 indicates habitat suitability stability, 1 indicates habitat suitability improvement, and −1 indicates worsening habitat suitability for only those areas with conditions previously considered "good".

The Max SSS threshold for each model was relatively consistent across model type, regardless of scenario (Table 2). The Low Correlation ($r < 0.5$) model had the lowest threshold value at 0.2662; all other models had a threshold value of at least 0.3.

**Prediction success of recent occurrence points**

Prediction success, or the percentage of occurrence points that the model correctly identifies as positive, was determined by the use of 20 additional occurrence points collected between 2001 and 2015 from museum records or directly from curators. These points were not used in the model constructions and thus provided an independent test of model results for data collected in between the two model timeframes (current and future). These points were displayed on a binary map of habitat suitability. Points that lay within the range of "suitable" habitat were predicted as true presences, while those that fell outside of the range were considered false negatives. Prediction success was then calculated by the equation $p.s. = a/(a+c)$ where $a =$ true positives and $c =$ false negatives. Additionally, we compared the logistic output at each site under both current and future climate models using a Wilcoxon signed-rank test.

## RESULTS

**Species distribution modeling—current climate**

No significant difference was found between the AUC for the four model types (ANOVA; $df = 3$, $p = 0.9253$, Fig. 3). All AUCs fell between 0.81 and 0.83 (Table 2), which is considered a good fit model (*Swets, 1988*). However, AUC values supplied by Maxent may differ from true AUC values because Maxent relies on background data and not true absence data, as noted in *Proosdij et al. (2016)*.

TSS values for all four model types fell between 0.5 and 0.65 (Table 2), which is considered a good fit model (*Landis & Koch, 1977*). The Full model had the strongest TSS score (0.6314), both Reduced and Moderate Correlation models had similar values (0.5831 and 0.5733), and the Low Correlation model had the smallest TSS value. The performance of the Low Correlation model is likely due to the limited number of variables included

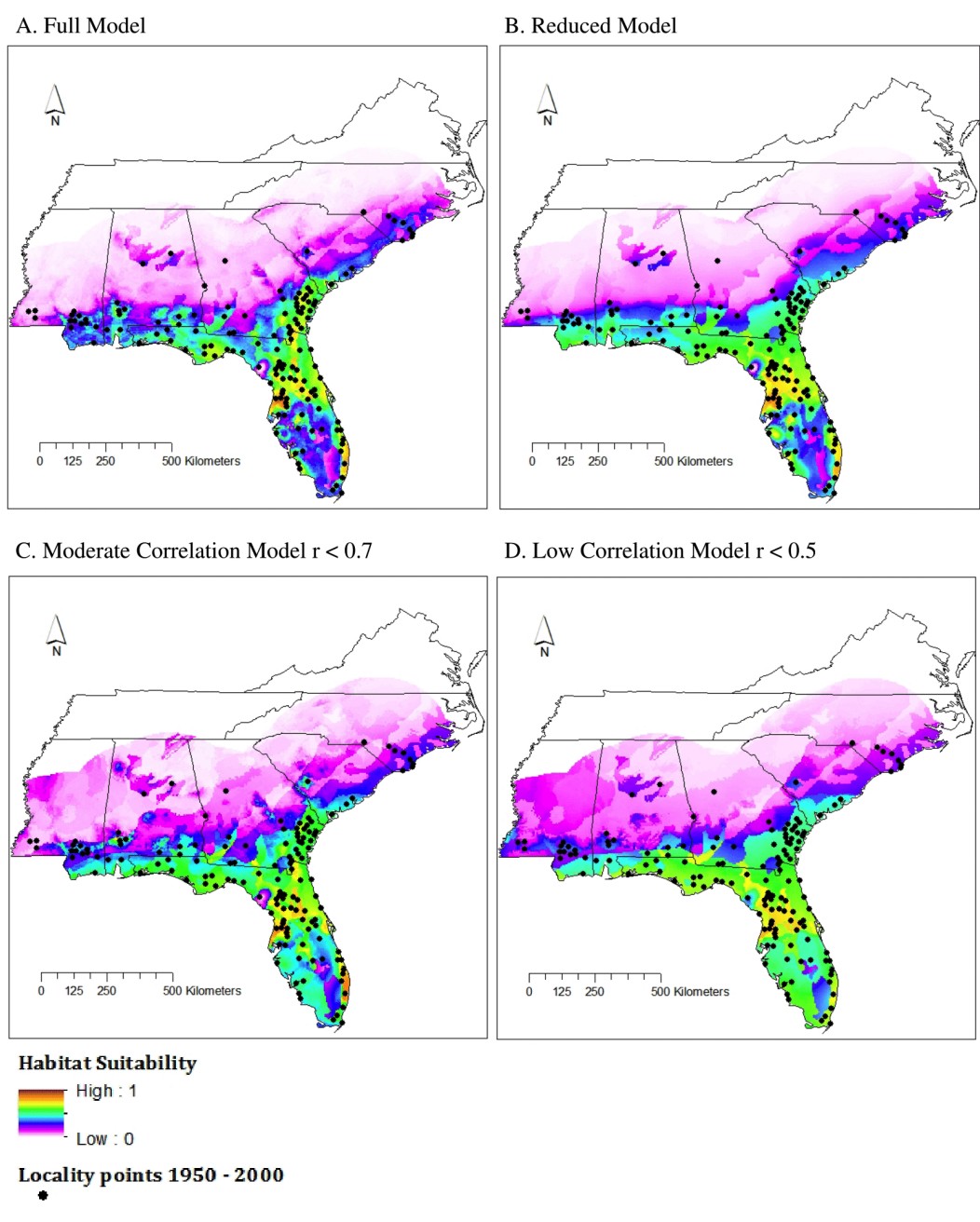

Figure 3 **Habitat suitability for *M. fulvius* throughout the Southeast under current climate conditions, 1950–2000.** (A) Full model, (B) Reduced model, (C) Moderate Correlation model, (D) Low Correlation model.

($n = 4$ of 20) as well as a lack of highly influential variables that were included in the Reduced model, which also had few variables ($n = 5$) but a higher TSS score.

All models predicted a range similar to that proposed by the IUCN (Fig. 3). The area of highest suitability was northcentral Florida. Using SDMtoolbox, we found high similarity in suitability predictions for all model types ($r < 0.9$); however, the Reduced model identified

a slightly greater extent of habitat suitability than the Full model. This is likely because the Reduced model was based off a small subset of the available environmental data (i.e., five out of the 20 variables because of our strict contribution requirements). Therefore, these models were the least constrained. As expected, the northern and central areas of Mississippi, Alabama, and Georgia were largely unsuitable aside from a small pocket of low suitability in central Alabama and south-central South Carolina. This small region in central Alabama is also highlighted by the IUCN range map. This is an area where *M. fulvius* has been known to occur historically, yet likely appears as only moderate to low suitability habitat in the models because we were only able to obtain two locality points from this region. Recent vouchered records of *M. fulvius* do not exist from this region in any of the databases we searched. Although this lack of recent records may stem, in part, from the fact that central Alabama remains one of the most herpetologically under surveyed regions in the southeastern United States, *M. fulvius* is known to be rare in this area (*Mount, 1975*).

In addition, western and central North Carolina were unsuitable areas for *M. fulvius*.

## Species distribution modeling—future climate

Under the GISS-E2-R (NINT) scenario, all model conditions gave similar results. Here, we show the results of the Full and Reduced models as those were the top two performing models. Both model types showed an increase in the logistic value of suitable habitat and a northward expansion of suitable habitat conditions (Fig. 4). The southern to central portions of Alabama, Georgia, and South Carolina displayed moderate to high suitability in the Full model but not the Reduced model. Both models showed areas of unsuitability in central Mississippi, parts of west-central Alabama, northern Georgia, and western and central North Carolina.

As seen in the GISS-E2-R (NINT) models, both MIROC-ESM scenario models also showed an increase in suitable habitat and a northward expansion of suitable habitat conditions (Fig. 4). In both model types, nearly all of Florida was considered to be moderately to highly suitable habitat and the majority of the northern boundary of our study region was predicted to be unsuitable.

## Climate novelty

MESS analysis identified multiple areas where no-analog or novel climates were present, which varied among models, years, and model types (Fig. 5). MIROC-ESM 2050 predicted a high degree of climate similarity in two of the four models (Low and Reduced); the Full and Moderate models, however, displayed almost no climate similarity. This would indicate that one or more of the future climate layers used in only these models may fall outside of the ranges seen in current times (1950–2000). The results from MIROC-ESM are similar to those obtained with GISS-E2-R (NINT) models (Fig. 5), where the Reduced and Low Correlation models show a higher degree of climate similarity than the Full and Moderate models.

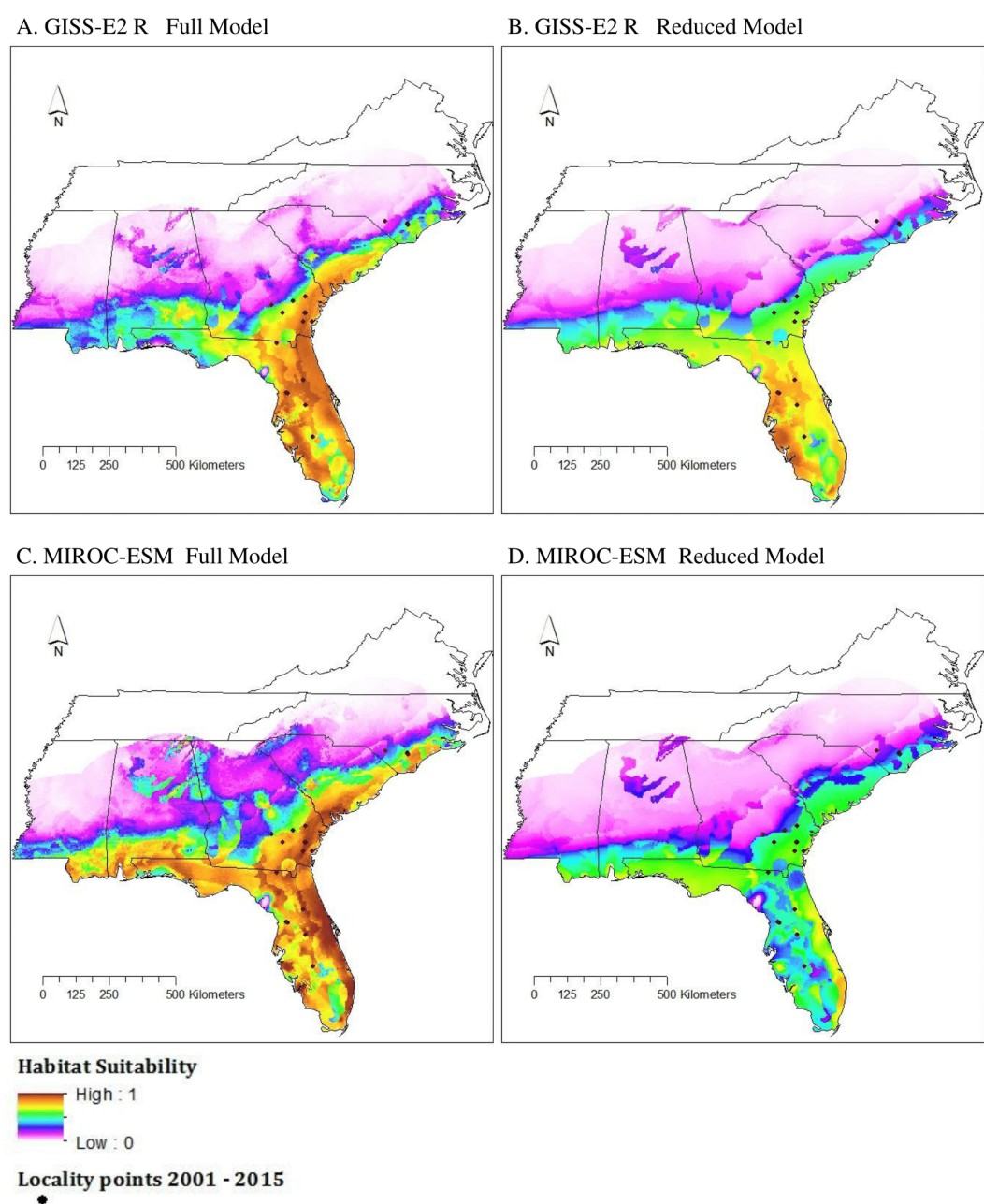

**Figure 4** **Habitat suitability for *M. fulvius* in the near future (2050) throughout the Southeast according to GISS-E2-R (NINT) ((A) Full model, (B) Reduced model) and MIROC-ESM ((C) Full model, (D) Reduced model).**

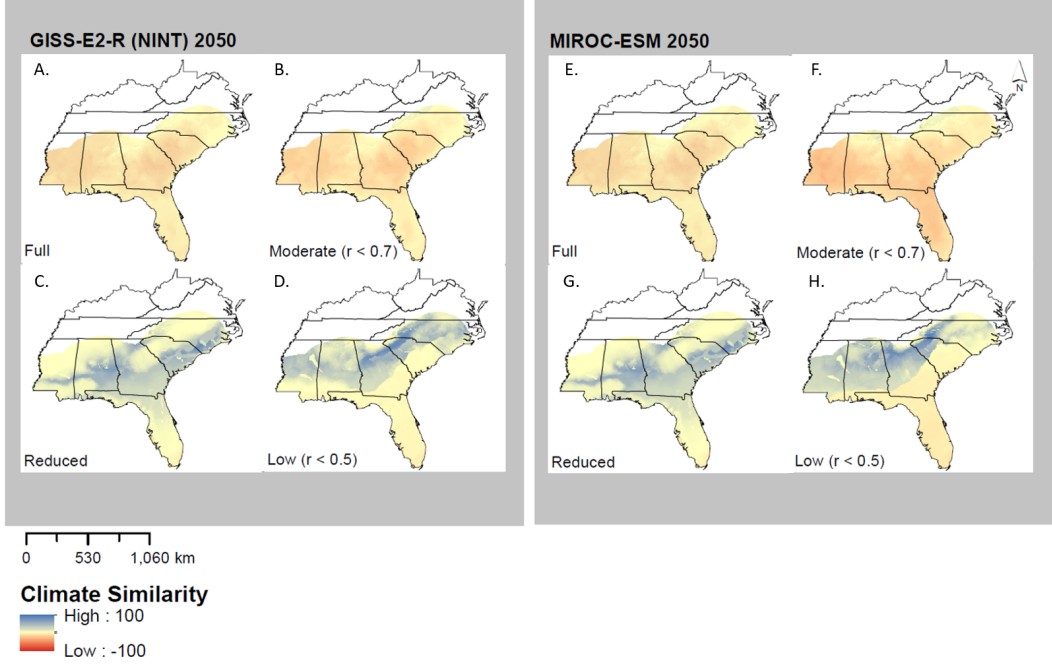

**Figure 5** **Multivariate environmental similarity surfaces (MESS) analysis for both GISS-E2-R (NINT) ((A) Full model, (B) Moderate correlation, (C) Reduced model, and (D) Low correlation) and MIROC-ESM ((E) Full model, (F) Moderate Correlation, (G) Reduced Correlation, and (H) Low Correlation) models.** MESS analysis measures climate similarity to training range when projecting a model. Negative values indicate a low similarity and therefore high climate novelty, while positive values indicate a high similarity and therefore low novelty.

## Model change

All models showed a northward expansion of various degrees from current conditions to 2050, consistent with our hypothesis (Fig. 6). Among the MIROC-ESM models, the Reduced model was the most conservative with predicting expansion. Similar results were found among GISS-E2-R (NINT) models. Areas of retraction were minimal and outweighed by areas of expansion.

As mentioned above, the Full model had the highest TSS score while the AUC scores were nearly identical across all four models. However, based on the effects of clamping and the MESS analysis, the Reduced model type is less biased by no-analog climate. The difference between the results of these two model types leads to slightly different predictions: the Reduced model predicts less range expansion than the Full model and is therefore more conservative.

## Description of suitable environment

Based on the differential results in identifying central peninsular Florida as suitable as well as the jack-knifing results, we predict that Temperature Seasonality, Mean Temperature of Coldest Quarter, Precipitation of Driest Month, Precipitation of Warmest Quarter, or some combination therein is highly influential for determining habitat suitability. Soil type

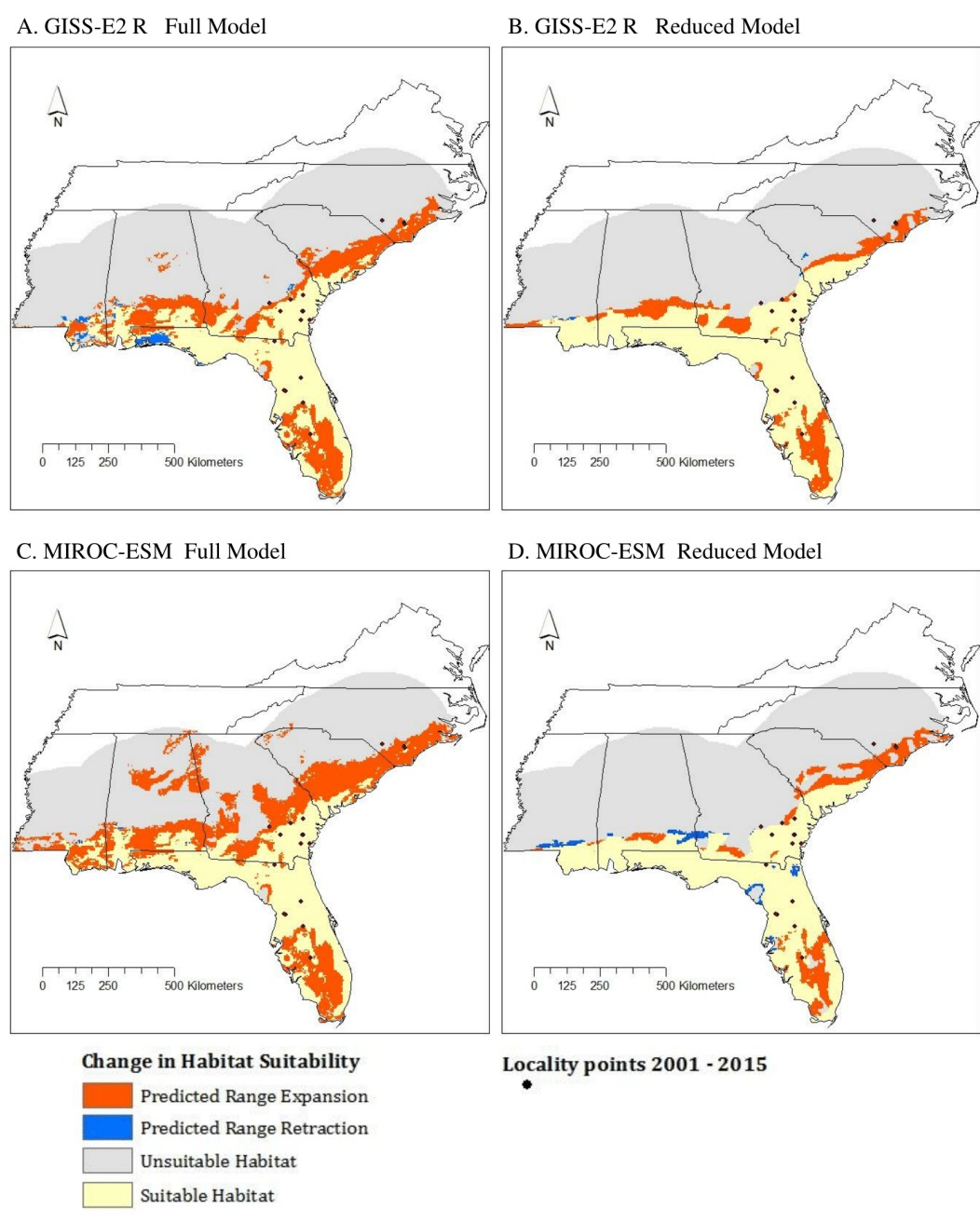

**Figure 6** Change in suitable habitat from current conditions (1950–2000) to near future conditions (2050) according to results of GISS-E2-R (NINT) ((A) Full model, (B) Reduced model) and MIROC-ESM ((C) Full model, (D) Reduced model) scenarios.

also had a significant influence on the output of the models. Because *M. fulvius* tends to inhabit areas of longleaf pine and scrub oaks, soil types that foster such vegetation would likely indicate more suitable habitat.

### Prediction success of recent occurrence points

The ability of the current climate models (1950–2000) to predict recently collected records ($n = 20$, collected between 2001 and 2015) was moderate across all four model types ($0.5 < p.s. < 0.75$). However, the 2050 models—both MIROC-ESM and GISS-E2-R (NINT)—had much higher prediction success ($0.8 < p.s. = 1$). Further comparing the logistic output at each site under current and 2050 conditions showed that there was a significant increase in habitat suitability under future climate conditions (MIROC-ESM 2050: $n = 20$, $W = 2$, $p < 0.001$; GISS-E2-R (NINT) 2050: $n = 20$, $W = 2$, $p < 0.001$).

## DISCUSSION

Current Maxent models suggest coral snake distributions are restricted to the southeastern United States, with only very limited areas of high habitat suitability; the majority of high to moderate habitat suitability areas are restricted to Florida. These results align well with the current known distribution of coral snakes (*NatureServe, International Union for Conservation of Nature, 2007*). Also, all four current climate models show relatively consistent geographic predictions. The largest region of difference between the models is in the southernmost tip of Florida which is shown as low habitat suitability by the Full model but more moderate suitability by the Low Correlation models; the other two models were intermediate between the Full and Low Correlation models.

Future climate models show a slight increase in habitat suitability throughout areas identified as moderate habitat suitability by the current climate models, and a slight expansion of suitable habitat along coastal regions. Both the GISS-E2-R (NINT) and MIROC-ESM models show similar geographic patterns. The Full models for both future climate change scenarios show more extensive range expansion than Reduced models. For the Full models, MIROC-ESM showed slightly greater range expansion than the equivalent GISS-E2_R (NINT) models; the extent of range expansion was more similar between MIROC-ESM and GISS-E2_R (NINT) for the Reduced models (Fig. 6). It is important to note that both the GISS-E2_R and MIROC-ESM climate data assume some global mitigation efforts will slow $CO_2$ emissions (*Kopp et al., 2014*). If no mitigation efforts occur, this RCP undoubtedly underestimates warming. As ectotherms are highly reliant on temperature, climate change is predicted to increase the potential distribution of many temperate reptile species in the Northern Hemisphere (*Currie, 2001*; *Hansen et al., 2001*; *Araújo, Thuiller & Pearson, 2006*). In particular, as *M. fulvius* already occurs at relatively high abundances in sub-tropical Florida, the increase in predicted habitat suitability in much of the Southeast as it warms is not surprising.

These results also, however, indicate that much of the Southeast may be well outside the range of current climate conditions, suggesting the presence of no-analog or novel environments within the Southeast in the future. Although habitats across the Southeast will warm, the extent of warming in the future will likely exceed recent maximum temperature

conditions (Fig. 5). Thus, these results must be interpreted with caution. These results are especially relevant as many reptile species are often not capable of optimal performance at the high end of their thermal tolerance and perform best at a lower temperature known as the "thermal optimum" (*Clusellas-Trullas, Blackburn & Chown, 2011*). It is unlikely that the upper limit will be reached, but it is possible that the thermal optimum may be exceeded which may have significant impacts on the behavior of reptile species. Although thermal data are lacking on wild individuals of *M. fulvius*, their fossorial tendencies and diel activity patterns (early morning and late evening) suggest that their thermal optimum could likely be exceeded by future warming in several areas of their range. Better physiological data that would allow an integration of mechanistic models with these correlative models (e.g., *Ceia-Hasse et al., 2014*) is clearly needed.

While climate change is often assumed to be a future issue, in reality it is likely already mediating range shifts and other potentially adverse species responses (*Parmesan & Yohe, 2003*; *Chen et al., 2011*). The increased prediction success of the future 2050 models, both MIROC-ESM and GISS-E2-R (NINT), indicates that climate may already be changing, resulting in range shifts in *M. fulvius*. This work suggests that studies using Maxent to make conservation decisions should strongly consider using future climate models in addition to models of the current climate conditions. This is particularly important for predicting the poleward range limits for ectothermic species that may already be undergoing significant range shifts, as these limits are often dependent on abiotic factors (*Cunningham et al., 2016*). Temperature and precipitation are the most common variables included in Maxent models, and most of the 19 bioclimatic variables have been used in more than 1,000 published models to date (reviewed by *Bradie & Leung, 2017*). These models may be misleading if only current climate is considered. Thus, we encourage future studies to include future climate projections as these predicted climate layers may improve model performance over using only current climate data.

Although the impact of climate change on shifts in species distributions has now been well documented, the impacts of climate change on the distribution of reptiles, and particularly snakes, remain understudied (but see *Araújo, Thuiller & Pearson, 2006*). Given that climate conditions from 2050 already better predict the distribution of *M. fulvius* than 'current' climate conditions, systematic surveys at northern range edges for *M. fulvius* and other snake species will likely yield valuable information on the rate of range shifts in response to climate change.

This work has important conservation implications for *M. fulvius. M. fulvius* is already infrequently encountered throughout its range and is extremely rare along its northern range limit. In North and South Carolina, where *M. fulvius* is currently at the range limit, habitat is expected to increase in suitability. Yet, high habitat fragmentation across the southeastern United States may limit the ability of this species to recover in these states where it is of conservation concern. The growing market for wood pellets in Europe could threaten US longleaf pine forests (*Tarr et al., 2017*), the preferred forest type of *M. fulvius*. In addition, an increasing demand for pine straw within the United States has left much of the longleaf pine forest that remains devoid of the leaf litter layer on which many fossorial reptiles, such as *M. fulvius*, depend. Thus, dispersal ability will be critical in driving whether

or not any species will be capable of occupying newly suitable habitat (*Araújo, Thuiller & Pearson, 2006*). In addition to habitat fragmentation, habitat preferences for environmental axes not included here may also limit this species' ability to respond as predicted to climate change.

Range expansion in *M. fulvius* (the venomous 'model') may also have important evolutionary consequences for two species of putative Batesian mimic: the scarlet kingsnake (*Lampropeltis elapsoides*) and the scarlet snake (*Cemophora coccinea*) (Fig. 1). Both mimic species occur well beyond the current range of *M. fulvius*. In allopatry, both mimic species have more red and less black on their dorsum than *M. fulvius* (*Harper & Pfennig, 2007*; *Akcali & Pfennig, 2017*), making them poor mimics. However, this breakdown of mimicry in allopatry is favored by predator-mediated selection: clay replicas of snakes that had similar amounts of red and black as *M. fulvius* were attacked more often in allopatry than the local phenotype with more red (*Pfennig et al., 2007*). In contrast, mimics that co-occur with *M. fulvius* at the edge of its range (e.g., in North Carolina) resemble it more precisely; in edge sympatry, there were fewer attacks on this phenotype than the redder allopatric one (*Harper & Pfennig, 2007*; *Kikuchi & Pfennig, 2010*). Thus, the migration of *M. fulvius* into previously uninhabited regions should facilitate the evolution of precise mimicry among local mimics. How rapidly precise mimicry evolves will depend on such factors as the generation times of predators and mimics, the standing variation in color pattern among mimics, the extent of gene flow between mimics from historical sympatry and mimics from 'novel' sympatry, and the strength of selection for precise mimics.

## ACKNOWLEDGEMENTS

The authors would like to thank all curators whose published data were used for this research. Additional thanks goes to Jeff Beane, Herpetology Collections Manager of the North Carolina Museum of Natural Sciences, as well as the Georgia Museum of Natural History and the Mississippi Museum of Natural History for providing additional non-indexed data for this research. Finally, the authors would like to thank Michael McQuillan, Gregory Haenel, and David Vandermast for their comments on the manuscript.

### Funding
This work was supported by NIH funding to David Kikuchi (NIH-2K12GM000708-16) and the Elon Honors Program to Jennifer N. Archis. The funders had no role in study design, data collection and analysis, decision to publish, or preparation of the manuscript.

### Grant Disclosures
The following grant information was disclosed by the authors:
NIH: NIH-2K12GM000708-16.
Elon Honors Program.

## Competing Interests

The authors declare there are no competing interests.

## Author Contributions

- Jennifer N. Archis and Amanda J. Chunco conceived and designed the experiments, performed the experiments, analyzed the data, prepared figures and/or tables, authored or reviewed drafts of the paper, approved the final draft.
- Christopher Akcali performed the experiments, contributed reagents/materials/analysis tools, authored or reviewed drafts of the paper, approved the final draft.
- Bryan L. Stuart conceived and designed the experiments, contributed reagents/materials/analysis tools, authored or reviewed drafts of the paper, approved the final draft.
- David Kikuchi conceived and designed the experiments, performed the experiments, analyzed the data, contributed reagents/materials/analysis tools, authored or reviewed drafts of the paper, approved the final draft.

## Data Availability

The two files containing the species spatial data used in this work have been provided as Supplemental Files.

## Supplemental Information

Supplemental information for this article can be found online at http://dx.doi.org/10.7717/peerj.4647#supplemental-information.

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
