# Peer review of "Is the future already here? The impact of climate change on the distribution of the eastern coral snake (Micrurus fulvius)"

_PeerJ, doi:10.7717/peerj.4647_

## Round 0.1 · original submission · Major Revisions

Please in particular provide more complete details of the methods and uncertainties around the models and model results, as this is mentioned by all referees. Then reconsider and flow and context of the paper as referee 3 suggests, and more carefully proof read everything..

·

Basic reporting

No comment

Experimental design

From line 216 to 224
It is better if you could give the values for the following MaxEnt parameters which you have used in your model
1) Regularization multiplier
2) Maximum iterations

Validity of the findings

No comment

Reviewer 2 ·

Basic reporting

The manuscript utilizes a popular method to study climate driven range shift pattern of snakes in the US. This is an important contribution considering not much work has been done on these specific reptiles using this method in the past. The manuscript is written well and included methodology which utilizes established methods for assessing model accuracy for present and future conditions.

My specific comments on the sections listed below in sections with lines numbers

Line 105: Language could be improved to ensure the audience can understand the text, for example in this line, do the authors mean “evaluating the utility of future climate models in relation to recent coral snake occurrence prediction in the years between 2001-2015”?

Experimental design

Line 108-109 & 216-224: Model environment settings were not mentioned (i.e. Authors did not mention whether they used default settings of MaxEnt or modified some of the parameters and whether it was executed trough the java applet or through R).

Line 149: Although it was explained the utility of the buffer but nothing mentioned about why 200 km buffer was generated not 100 km or 50km? More details about this would be helpful for the readers.


Line 221: In this type of modeling studies it is expected the model repeated at least 100 times for ensuring the robustness of the result. Also bootstrapping in Maxent with observation data split into training and test run is often used for the replicate runs to add additional layer of validation trough AUC’s. Authors need provide justification why model was only repeated 10 times and without any data split.

Validity of the findings

no comment

Additional comments

Additional figures could be added in the appendix showing all the environmental layers used for the modeling which will help readers to understand whether the distribution could be derived by particular environmental variable or with combination of them.

I thank the authors for providing the raw data but including a descriptive metadata would be useful for the readers and also raw data of ‘locality points 2000-2015’ file need to have ‘Species’,’ data source’ fields like other location file.

Authors could consider the following citations to enhance their introduction and discussion section
Ceia‐Hasse, Ana, et al. "Integrating ecophysiological models into species distribution projections of European reptile range shifts in response to climate change." Ecography 37.7 (2014): 679-688.

Araújo, Miguel B., Wilfried Thuiller, and Richard G. Pearson. "Climate warming and the decline of amphibians and reptiles in Europe." Journal of biogeography 33.10 (2006): 1712-1728.

Chen, I-Ching, et al. "Rapid range shifts of species associated with high levels of climate warming." Science 333.6045 (2011): 1024-1026.

Reviewer 3 ·

Basic reporting

This paper studied the present and future distributions in the eastern coral snake (Micrurus fulvius) in the United States. They also compared the current and future distribution models to see if the distribution range shifts are already happening in this specie or not.

This paper has some important results to science where there is a gap of knowledge on climate change impacts on future distributions on reptiles. However, I believe that this paper in its current format is not publishable and needs careful and considerable revisions. Please find my comments in details as follow and also in the attached annotated PDF file.

Experimental design

Introduction
The structure of the paragraphs are not following each other. Most of the paragraphs do not have a main topic and are just some general introduction.
Although you mentioned three objectives for your research, but still the importance of doing this research and the benefit of the outcomes are not clear enough. You need to mention what is the main gap of knowledge and how your study addressed this. What would be the benefit of your future models to establish management plans for the areas of future inductions and so on.
You also need to talk about the physiology, ecology, and habitat preferences of the coral snakes. These information are missing in the discussion as well.

Methods
Some methods which have been used in this study need to be explained more in details in the Methods and also the Methods need to be shortened and be more concise. I mentioned some examples both in the annotated PDF and as follow:

1- you need to explain why you chose these environmental factors. You should explain clearly the role and importance of each layer in biology, physiology, or ecology of the coral snakes.
2- you should add a specific part to your method explaining the most important environmental factors in creating your present and distribution models. This part is very important to explain the models based on the contribution and importance rates of the layers you used.
3- you have to include more information for the data, e.g., Unit, Type, Temporal range, Minimum, Maximum, Mean, Std. Dev.

Validity of the findings

I would like to see the response curves or at least a table which shows the contribution and importance rate of each layer in both present and future models. This part is really missing from the paper and it is one of the most important outcomes of the Maxent modeling.

Additional comments

1- please use more recently studies on climate change modelling using Maxent and compare your result with those ones. There are many studies have been done since 2014 which many of them are missing from your reference list.
2- please minimize the use of one-off reference in your general statements. Once you are talking a bout a fact and staying many studied found it, you should at least give 2-3 references, please refer to the annotated PDF.
3- please also divide the discussion in two parts, present and future modeling. In the current format, discussion is not concise and very hard to follow what you mean in many of the paragraphs. At the end of the Discussion, I was left with a "so what" question.
4- please revise the formatting of the reference list, there are many typos, e.g., lines 550 (issue No. in Bold), 668 (year in Bracket), and 729 (issue No. is missing).

---

## Round 0.2 · Minor Revisions

The referees have given the paper another very detailed review and have some suggesitons that will improve the paper. Notably correcting a corrupted figure in an appendix and colours in figures (less colours is always preferable as many people are partly colour blind and colours never reproduce the same way as in the originals). They also ask for clarifications in methods and amendments to clarify text elsewhere. Please attend to the carefully. Regarding referee #3 question, I believe common names of species are not capitalised (unless they are a person or place name); so your text is correct already.

·

Basic reporting

This manuscript is a thorough study of the impact of climate change on the distribution of the eastern coral snake (Micrurus fulvius). MaxEnt (version 3.3.3. k) species distribution modeling software was used to evaluate range shifts from 1950 to 2000 and 2050 of the eastern coral snake. The authors report that the future climatic conditions will increase the habitat suitability across the range. They predict the Southeastern United States will have a novel environmental condition in 2050. This paper has important findings for reptile conservation and in biogeographical studies.

Experimental design

Thank you for making changes in line 220-222.
The study has been carefully designed and the steps of the methods are clear and easy to follow. Nevertheless, I have two additional general comments:
1) Line 160 & 277: The conversion of arcminutes to km. The given conversion is incorrect. It must be approximately 4.6 km at the equator.
60 arcminutes = 1 degree = approximately 111 km at the equator.
5 arcminutes = 9.2 km at the equator.
2) Line 220 -222.
There are many successful studies which have used cross-validation and 10 replicates. Please add references and provide a basic description of cross-validation.

Validity of the findings

Thank you for providing supplementary files.
It is difficult to see whether the resulting MaxEnt model maps have a higher prediction for the true natural habitats (true presence) from 1950 to 2000.
I would like to suggest to change the color of the maps in figure 2. It is better if you can reclassify these resulting maps using a mapping software and give contrast colors for each class. After that add occurrence records to the same map for the better visualization.

Additional comments

No comment

Reviewer 2 ·

Basic reporting

no comment

Experimental design

Thanks for implementing suggested revisions in the methods, the strategy followed to generate the model is much clearer now compared to the previous edition.

Validity of the findings

no comment

Additional comments

I thank the authors for the successful revision of the MS by following the suggested revisions by the reviewers, the article did improve a lot. The response curve word document in Appendix 3 is corrupt, please fix the document with the correct version.

Reviewer 4 ·

Basic reporting

Overall this paper is a pleasure to read. The authors are correct that little distribution work, with modeled or not, has included snakes, particularly a secretive but recognizable one such as the Eastern Coral Snake, so this paper is a solid contribution in that regard. Their approach is sufficiently well constructed and thoughtful to be a model not only for other snake species but other under-studied vertebrates with sufficient field occurrence data. Specifically matching occurrence years with modeled climate years and examining distributions with respect to future climate modeling (including the bracketing of projections) will be important in our rapidly changing world.

The authors do not discuss much of the natural history of these secretive snakes, which I view as both a challenge for modeling and also makes them good candidates since models can extrapolate distributions from limited sources. The discussion of the paper would feel more “complete” if they discuss their modeling in light of the natural history of Micrurus fulvius and how that may impact SDMs and their interpretation.

Their most interesting conclusion is that they may have shown with their future climate models that the snakes are already responding to climate change. They could improve their case with some easy fixes to figures (see below). Impact on Batesian mimic species, both the kingsnake and scarlet snake, is a fascinating discussion topic, and I am glad they included this.

Generally, the authors use available occurrence and climate datasets and methods that this is reproducible especially given the submitted locality sites used in the study. I note the site data is only sites; no specific specimen information like GUID is provided which would make it difficult to verify any voucher specimens, but the authors make no claims to have done so either.

The authors responded thoroughly and appropriately to previous reviewer comments to which I had access.

Experimental design

Three main goals are stated:
Line 19: "Specifically, we: (1) identified the current range and suitable environment of M. fulvius in the Southeastern United States, (2) investigated the potential impacts of climate change on the distribution of M. fulvius, and (3) evaluated the utility of future models in predicting recent (2001-2015) records. "

Methods
SDM methods are explained well and are reasonable and justified; however, it’s confusing to have the authors refer to the sites in the SDM section of methods before talking about the occurrence data. I would suggest swapping the order. In other words, describe the locality data from museums, etc first then the environmental data and spatial extent since it is in part based on those localities.

They chose to use Maxent using k-folding (10 reps) which is one way of mitigating inherent model bias. My biggest issue while reading this initially was why the four parameterized versions of the current climate? Most studies caution against using all available variables to avoid overfitting in SDMs- the authors point this out (it appears that way in their Fig 3A) so I was expecting more discourse on methodology in the results. They do address these differences in Results. It would have been expected that they select the most robust model balancing model performance statistics and extent of suitable habitat for the rest of the analysis and discussion, which in my opinion is the Reduced model. They chose to bracket their modelling with the Full model and Reduced model for the future climate models, which is acceptable.

Environmental layers include WorldClim 1.4, Soils (FAO), justifying their choice of resolution at 2.5 arcmin based on the uncertainty estimates of the species occurrence.

Line 184: a little description of the SDMToolbox would be helpful for readers who may not know about this set of python scripts for ArcGIS 10 (especially given the level of detail they went into for other aspects of their Methods). Please add the proper reference (Brown et al 2017). They do supply a weblink.

Line 190-191: references for “Maxent’s minimum location requirement” As far as I know Maxent itself has no minimum threshold but there are many references to what are effective minimum sample sizes, including Graham et al 2006 and other references already used in the paper to highlight some of the discussion on this topic. Could this be reworded to be less misleading?

Line 236: nice summary of GCMs and RCPs. They choose a 4.5 RCP (low emissions scenario) — are they underestimating climate impact? They bracket a high and low future projections by choosing MIROC-ESM and GISS-E2-R but still may be underestimations.

They tested model performance with AUC and TSS (Cohen’s Kappa) as well as used a Wilcoxon signed-rank tests on the thresholded models (current and future) with an independent set of samples. These are conventional and acceptable methods.

Validity of the findings

With respect to the authors stated goals (listed in same order as Line 19):
1) this was done with four different parameterized approaches for current climate suitability. While the authors point out the general agreement of the models, there is variability in the southernmost extent in Florida. I wonder why all the current models show little/no suitability for the isolated northernmost portion of its range in Alabama? - the authors mentioned that in Alabama, the species is of high conservation concern. Are the climate models at this resolution failing to capture some local phenomena or is the occurrence of the snake there questionable? Or an issue with the SDMs? Perhaps it’s out of scope to delve into the factors thoroughly but should be worth noting in results or discussion since we are presented with the IUCN range map and then their distribution models, and it is relevant to their case that there is predicted expansion.
2) climate change models show much expansion and in some ways appear more like the IUCN range maps (which granted come with a large grain of salt) than the current climate models (also noted by authors). There is enough concordance between the two CMIP5 models that the overall future projections are convincing. However they do not address the fact that they are undoubtedly underestimating climate shifts with their RCP 4.5 even bracketing a pessimistic and more gentle models of change.
3) this is the most interesting (and speculative) part to me, and they conclude with caveats that the most recent localities which are predicted in the future climate models indicate that there are climate change responses already. I would prefer they visualized these data to be more convincing. (see note below)

Nice use of MESS analysis to identify no-analog/ novel climate space in the future climate models. It is valuable to see the differences based on model treatments and the CMIP5 models themselves.

Line 363 "The Full model type was presumed the most valid of the four models based on TSS and AUC” — but not by much as AUC values range from 0.828 to 0.8148 and had similar TSS. I think the Full model performed the best but I do not assume it’s the most “valid” which implies that it’s closest to the actual distribution of the species. As I mentioned before under Methods, weighing in the differences in extent of suitable habitat, the MESS, I favor the Reduced model but I am ok with a difference of opinion. I would prefer this Line reworded.

Tables and Figures, when checked how they were referenced in text, are in order and appropriate. Is there any figure or supplemental data that show testing vs. training data? Also, I would find it helpful, that is strengthen your case, to see the occurrence points on all the climate models, especially the future models since part of their conclusion that the species is already undergoing shifts in response to climate change is that the models better match the more recent localities. That is a big statement that would be best supported with a figure (either Fig 4 or 6, even if included in supplemental).

Figure 6: the color for Retraction is too similar to Expansion; please change (maybe red?)

Line 407: I appreciate this cautioning of interpreting modeling with no-analog climate; they reference appropriate papers here and discuss thermal optimum as a limitation. Does a largely fossorial species have special consideration here?

Line 448-449 I am glad they included this observation on dispersal limits as being also a limitation of this study. It would be worthwhile to include some more species specific details, such as habitat preference of forest types and understory/ leaf litter lifestyle and how these correlate with trends in forest loss or gain in the SE-US (an area which has seen much change in pine forest composition) and their models. (Tip: Global Forest Watch www.globalforestwatch.org has an interactive map that shows a lot of forest turnover in the SE.)

Additional comments

Title: Are not recognized common names of species capitalized? If so, it should be Eastern Coral Snake

Details to clean up:
Line 64 vs Line 67 - what are PeerJ guidelines? "Fig. 1 “ versus "Fig 1”? Or spelled out? Should be fixed to be consistent

Line 159: Typo in genus name “Mircrus” (sic)

Appendix S2 heading: please capitalize consistent with the organizations’ preference and your Methods section (ie., GBIF, VertNet, iDigBio)

Supplemental figures (all the bioclimatic maps) have two north arrows (likely unintentional) that should be cleaned up.
The response curves did not have headings and my review copy had a corrupted Word Doc so it was difficult to know what I was looking. Which models were these the response curves for?

---

## Round 0.3 · accepted · Accept

Thank you for the careful revisions and improved figures for what is an interesting and well presented paper. If there is not special reason to have so many SM files I would include all the text ones in one file to make it easier for readers to download fewer file; but that is your discretion.

#